# New Species and Records of *Lemophagus* Townes, 1965 (Hymenoptera, Ichneumonidae, Campopleginae), from China [note 1]

**DOI:** 10.3390/insects15120932

**Published:** 2024-11-27

**Authors:** Yuanyuan Han, Chengxue Wei, Chong Liu, Yan Dong

**Affiliations:** 1College of Biology and Food Engineering, Chuzhou University, Chuzhou 239000, China; yyhan6@zju.edu.cn (Y.H.); wcx19955805965@126.com (C.W.); lcindividual@126.com (C.L.); 2School of Life Sciences, Anhui University, Hefei 230000, China

**Keywords:** *Lemophagus*, new species, new record, morphology, taxonomy

## Abstract

The genus *Lemophagus* belongs to the subfamily Campopleginae and is mainly distributed in the Holarctic and Oriental regions. To date, only nine extant species have been recorded. The hosts of this genus are mainly the subfamily Criocerinae (Chrysomelidae). In this study, Chinese species of the genus *Lemophagus* were reviewed based on comparative morphological characteristics. In total, one new species and two new records from China are described.

## 1. Introduction

Campopleginae is a cosmopolitan subfamily of Ichneumonidae, which contains about 2200 species in more than 60 genera [1]. The members of this subfamily are mainly koinobiont endoparasitoids of Lepidoptera larvae [2,3,4,5,6,7,8], although some species attack Coleoptera [9,10], Diptera [11], Hymenoptera [12], Rhaphidioptera [13] and Neuroptera larvae [14]. Therefore, campoplegines are effective parasitoids for the biological control of pests [15,16,17].

The genus *Lemophagus* Townes, belonging to Campopleginae (Hymenoptera: Ichneumonidae), was established by Townes in 1965. *Lemophagus curtus* was designated as the type species of the genus by Townes in 1965. This is a small genus comprising nine species distributed in the Holarctic and Oriental regions [18,19,20,21]. Most of the species are known from the Holarctic region, except for one species, *L. japonicus*, which ranges from the Eastern Palaearctic to the Oriental region [20,22]. The identification of genus *Lemophagus*, which is very similar to the genus *Olesicampe*, is always not easy, and sometimes it can be confused with the genus *Hyposter* [23,24,25]. They share some intermingled characters, such as a hind wing with a nervellus that is not intercepted, a glymma on the petiole that is faintly indicated or entirely absent and an ovipositor about as long as the apical depth of the metasoma. However, it can be distinguished from them by the always irregular rugulose-to-rugose face, the conspicuously broad clypeus, the apical margin of the clypeus being slightly out-turned and proposed and the subapical notch on the dorsal valve of the ovipositor being unusually far from the apex (apical 0.3 of ovipositor) [23,24,25].

Species of *Lemophagus* are solitary endoparasitoids [26,27]. Unlike most campoplegine genera, they most commonly parasitize chrysomelid larvae living exposed on foliage [28,29]. Only *L. foersteri* was recorded from larvae of *Spodoptera exigua* [25,30]. Therefore, it is an important group for the control of agricultural pests [31]. A good example is provided by *L. curtus*, which has been released in the USA to control *Oulema melanopus* [32,33,34]. Recently, there have been a number of works on *lemophagus* species assessing their biocontrol potential released in the field, particularly against *lilioceris lilii* [31,35,36].

In this study, Chinese *Lemophagus* specimens conserved in the Biodiversity Resources Repository, Chuzhou University (CHZU), were examined. One new species and two new records are reported from China. 

## 2. Materials and Methods

This study is based on specimens preserved in the Biodiversity Resources Repository, Chuzhou University (CHZU).

The terminology and measurements used follow Broad et al. (2018). All description and measurements were made under ZEISS Stemi 305 microscopes, and all figures were made by digital microscope (VHX-8000C, KEYENCE, Osaka, Japan). Type specimens are deposited in the Biodiversity Resources Repository of Chuzhou University (Chuzhou, Anhui province, China).

## 3. Results

### 3.1. Taxonomy

Subfamily: Campopleginae Förster, 1869

Genus: *Lemophagus* Townes, 1965

Type species: *Lemophagus curtus* Townes; by original designation.

Diagnosis: Front wing 2.5 to 5.0 mm, body length 3.0 to 7.0 mm; clypeus conspicuous wide, and its apical margin out-turned and proposed; face granulose-rugulose to rugose-punctate; flange on lower margin of mandible narrow; temple strongly narrow behind eyes; nervulus opposite basal vein or slightly distad of basal wing; nervellus vertical, not intercepted; areolet present, rhomboid, petiolate above, emitting vein 2m-cu from its apical half; mesoscutum with notaulus absent; propodeum with area superomedia usually confluent with area petiolaris; glymma on petiole faintly indicated or entirely absent; dorsolateral carinae weakly to distinctly present; subapical notch on the dorsal valve of the ovipositor situated at apical 0.3 of the ovipositor (which is unusually far from the apex); ovipositor about as long as apical depth of metasoma; tarsal claw simple to strongly pectinate.

#### 3.1.1. *Lemophagus curtus* Townes, 1965 (Figure 1 and Figure 2)

Material examined: 1♀, Shanxi, Lishan, Lidan Zhang leg., 7.VIII.2022, No. 202306928; 8♀4♂, Anhui, Huangshan Mountain, Qingyuan Zhou leg., 4.VII.2024, No. 202402908, 202402929, 202408122, 202401186, 202402990, 202401182, 202403959, 202401185, 202401219, 202402139, 202402863, 202403354; 1♀, Yunnan, Yunlong County, Yellowpan Trap, 21.VIII.2023, No202306361; 10♀, Anhui, Xuancheng, Yuanyuan Han leg., 3.VII.2023, No. 202303515, 202302919, 202302830, 202303272, 202303434, 202305159, 202302747, 202302090, 202303309, 202300069; 1♀1♂, Henan, Gushi, Yicheng Ding leg., 24.VII.2024, No202403321, 202403314.

Description: Female (Figure 1; No. 202306928 from Shanxi province). Body length 3–5 mm in Horstmann, 2004, 4.8 mm in this specimen and fore wing length 3.7 mm.

**Figure 1 insects-15-00932-f001:**
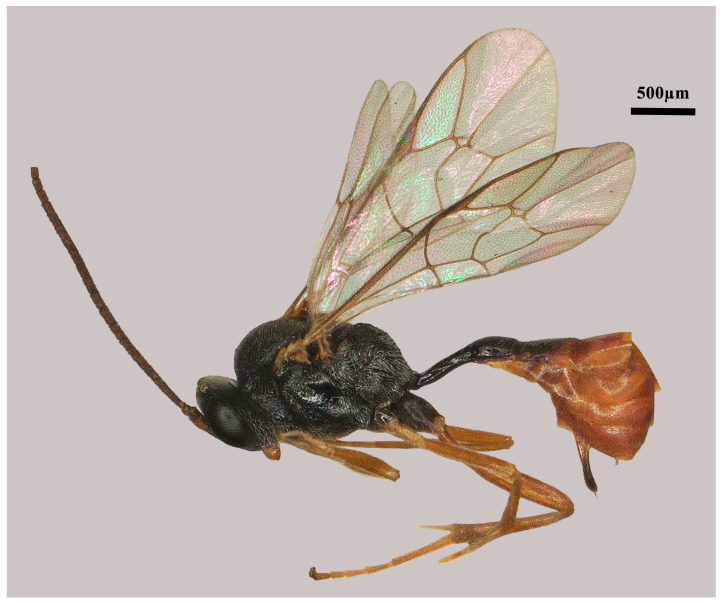
*Lemophagus curtus* Townes, 1965. No. 202306928 from Shanxi province, female, habitus.

Head: Antenna with 26 flagellomeres; first flagellomere ca. 3.2× longer than its apical width; preapical flagellomeres longer than wide; face (Figure 2E) rugose-punctate; clypeus very weakly separate from face, weakly punctate; malar space mat, ca. 0.6× as long as basal width of mandible; upper tooth of mandible slightly longer than lower tooth, with a weak lamella; frons rugose-punctate and with median carina; vertex (Figure 2F) mat; ocular-ocellar distance 1.3× longer than diameter of lateral ocellus, distance between lateral ocelli 1.6× longer than diameter of lateral ocellus; occipital carina complete, reaching hypostomal carina before base of mandible.

**Figure 2 insects-15-00932-f002:**
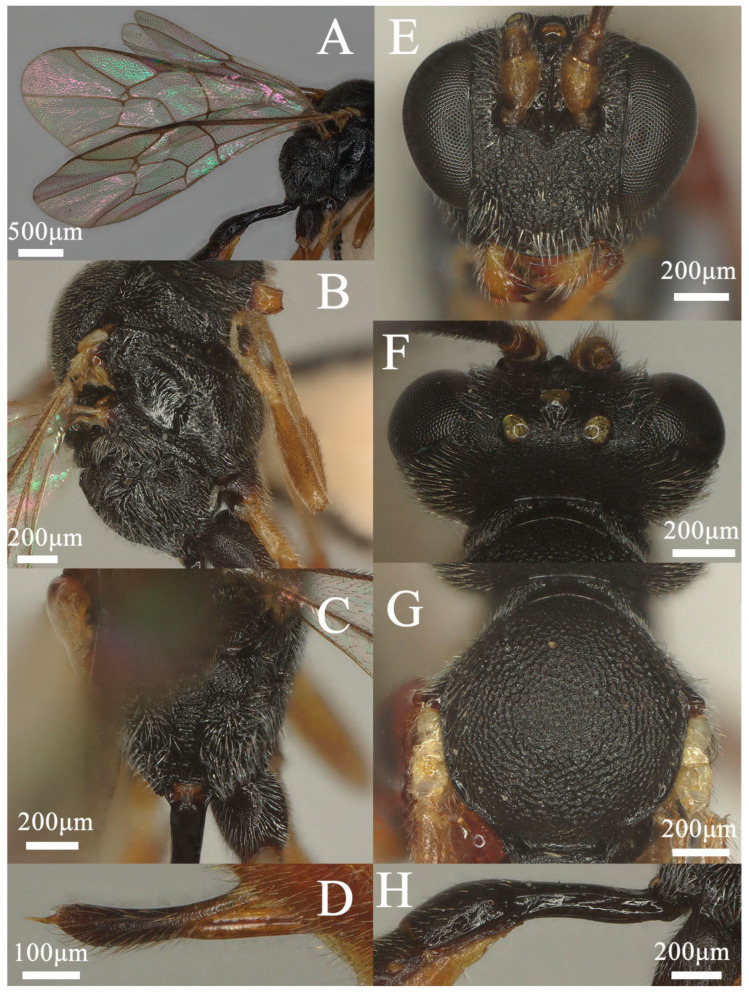
*Lemophagus curtus* Townes, 1965. No. 202306928 from Shanxi province, female. (**A**) Fore wing; (**B**) Mesosoma, lateral view; (**C**) Propodeum, dorsal view; (**D**) Ovipositor, lateral view; (**E**) Head, anterior view; (**F**) Head, dorsal view; (**G**) Mesoscutum, dorsal view; (**H**) First metasomal segment, lateral view.

Mesosoma: Pronotum rugose-punctate dorsally and transversely striate ventrally. Mescoscutum (Figure 2G) punctate, becoming rugose-punctate on notaulic area. Scutellum convex in profile, with lateral carinae basally. Scutellum and metanotum rugose-punctate. Mesopleuron (Figure 2B) punctate, trans-striate below tegula, speculum smooth and shiny, sternaulus present at basal 0.3; metapleuron punctate, sculpture becoming denser on juxtacoxal area, short rugose posteriorly; propodeum (Figure 2C and Figure 7A) with area basalis trapezoid; area superomedia large, hexagonal, densely rugose; area petiolaris irregularly rugose, not concave medially, confluent with area superomedia; area externa and area dentipara ruguse-punctate; area postero-externa rugose; area lateralis finely rugulose with shallow punctures; costula developed; lateral longitudinal carina developed; propodeal spiracle oval.

Wing: Fore wing (Figure 2A) areolet present emitting vein 2m-cu from its apical half, with a moderate stalk. Marginal cell short, distal part of surrounding vein 1.7× longer than proximal one. Vein 1cu-a opposite M&RS. External angles of second discal cell acute (70°). Hind wing with CU&cu-a vertical, not intercepted.

Legs: Hind femur ca. 4.7× longer than wide. First tarsal segment without ventral row of setae medially. Tarsal claws curved apically, pectinate.

Metasoma: First metasomal segment (Figure 2H) without glymma, dorsolateral carinae of first metasoma moderately developed. First tergite 2.5× longer than width of postpetiole. Second tergite 0.6× as long as first tergite, 0.9× longer than its apical width; thyridium elliptic and separated from base of second tergite by length of its length. Third tergite 0.9× as long as its apical width. Ovipositor (Figure 2D) nearly as long as apical depth of metasoma, with a notch on upper valve.

Color: Black. Scape and pedicel yellowish-brown; mandible yellow except teeth reddish-brown apically; tegulae whitish-yellow; all legs yellowish-brown except coxae blackish-brown and apical tarsus darker. First metasomal segment blackish except yellowish-white extreme apically; second and third tergites brownish basally, yellowish-brown apically; remainder tergites brownish dorsally and ventrally, yellowish-brown laterally.

Distribution: China (Anhui, Henan, Shanxi, Yunnan), Austria, Bulgaria, Czech Republic, Czechoslovakia, Denmark, France, Germany, Italy, North Korea, Poland, Romania, Russia, Sweden, Switzerland, Ukraine, United Kingdom, USA.

Variation: Antenna with 25–27 flagellomeres, first flagellomere 3.2–3.7× longer than its apical width; pronotum punctate to rugose-punctate dorsally; hind femur 4.7–5.0× longer than wide; area externa punctate to rugose-punctate; marginal cell with surrounding vein 1.5–1.9× longer than proximal one; first tergite 2.2–2.5× longer than width of postpetiole; second tergite 0.6–0.7× as long as first tergite, 0.8–0.9× longer than its apical width.

Remarks: This species is first reported here from China and the Oriental region.

#### 3.1.2. *Lemophagus pulcher* (Szépligeti, 1916) (Figure 3 and Figure 4)

Material examined: 4♀3♂, Jiangsu, Qionglong Mountain, Malaise trap, 2.VII.2022-4.VIII.2022, No202300005, 202300006, 202300229, 202301452, 202301069, 202301388, 202301578.

Description: Female (Figure 3; No. 202300005 from Jiangsu province). Body length 6–7 mm in Horstman, 2004, 9.1 mm in this specimen and fore wing length 6.2 mm.

**Figure 3 insects-15-00932-f003:**
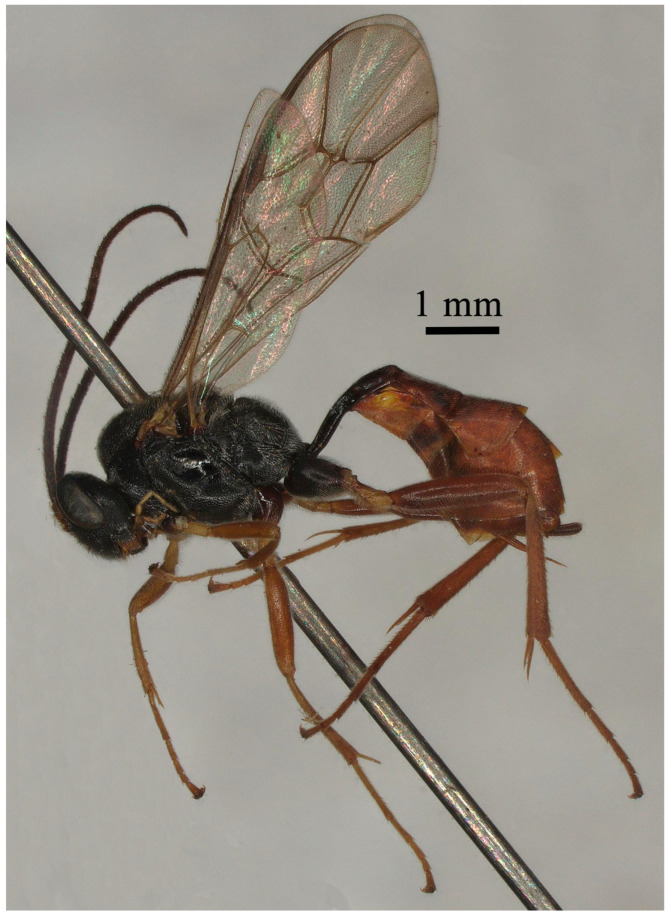
*Lemophagus pulcher* (Szépligeti, 1916). No. 202300005 from Jiangsu, female, habitus.

Head: Antenna with 32 flagellomeres; first flagellomere ca. 2.9× longer than its apical width; preapical flagellomeres longer than wide; face (Figure 4E) finely rugose; clypeus very weakly separate from face, rugose and becoming rugose-punctate apically; malar space finely rugose, 0.45× as long as basal width of mandible; upper tooth of mandible as long as lower tooth, with a weak lamella; vertex (Figure 4F) finely granulose; ocular-ocellar distance 1.6× longer than diameter of lateral ocellus, distance between lateral ocelli 2.3× longer than diameter of lateral ocellus; occipital carina complete, reaching hypostomal carina at base of mandible.

Mesosoma: Pronotum punctate to rugose-punctate and transversely striate ventrally, the distance between punctures shorter than its diameter. Mesoscutum (Figure 4G) punctate and becoming rugose-punctate on notaulic region. Scutellum convex in profile, with lateral carinae basally. Scutellum and metanotum rugose-punctate. Mesopleuron (Figure 4B) punctate, the distance between punctures at least as long as its diameter, trans-striate below tegula, speculum smooth and shiny, tending to be rugose above episternal scrobe; sternaulus present at basal 0.4; metapleuron smooth to indistinctly punctate on upper half, rugose-punctate at juxtacoxal area; propodeum (Figure 4C and Figure 7B) with area basalis trapezoid; area superomedia hexagonal, convergent behind costulae, posteriorly opened, sparsely rugose, as long as wide; area petiolaris trans-rugose, broadly concave, more or less separated from area superomedia; area externa rugose-punctate; area dentipara shallowly punctate and short irregularly rugose; area postero-externa irregularly rugose; area lateralis shallowly punctate with the trace of rugulosity along lateral carina; all carinae strongly developed; propodeal spiracle oval, connected to pleural carina, closer to lateral longitudinal carina than pleural carina.

Wing: Fore wing (Figure 4A) areolet present emitting vein 2m-cu from its apical half, with a long stalk. Marginal cell short, distal part of surrounding vein 1.6× longer than proximal one. Vein 1cu-a opposite M&RS. External angles of second discal cell acute (70°). Hind wing with CU&cu-a nearly vertical.

Legs: Hind femur ca. 4.3× longer than wide. First tarsal segment without ventral row of setae medially. Tarsal claws curved, strongly pectinate.

Metasoma: First metasomal segment (Figure 4H) without glymma, dorsolateral carinae of first metasoma strongly developed. First tergite 2.9× longer than width of postpetiole. Second tergite 0.5× as long as first tergite, 0.83× as long as its apical width; thyridium almost round and separated from base of second tergite by length of its diameter. Third tergite as long as its apical width. Ovipositor (Figure 4D) nearly straight, as long as apical depth of metasoma.

Color: Black. Scape basally and pedicel yellowish-brown; teeth medially yellowish-brown, reddish-brown apically and blackish-brown basally; fore and mid coxa blackish-brown, trochanter and trochantelli whitish-yellow, remainder of fore and legs yellowish-brown; hind coxa black, trochanter brownish basally and yellowish apically, trochantelli whitish-yellow, remainder of hind leg brownish; first metasomal segment black, remainder of metasoma yellowish-brown tinged with black mark.

Distribution: China (Jiangsu).

**Figure 4 insects-15-00932-f004:**
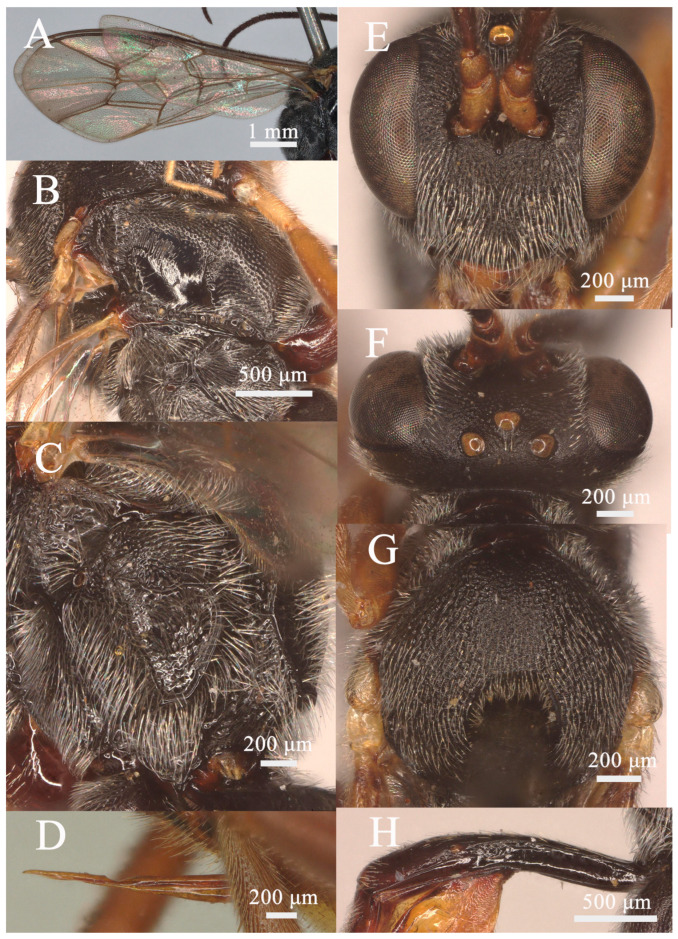
*Lemophagus pulcher* (Szépligeti, 1916). No. 202300005 from Jiangsu, female. (**A**) Fore wing; (**B**) Mesosoma, lateral view; (**C**) Propodeum, dorsal view; (**D**) Ovipositor, lateral view; (**E**) Head, anterior view; (**F**) Head, dorsal view; (**G**) Mesoscutum, dorsal view; (**H**) First metasomal segment, lateral view.

#### 3.1.3. *Lemophagus nanus* sp. nov. (Figure 5 and Figure 6)

Material examined: Holotype: ♀, Anhui, Langya Mountain, Yuanyuan Han leg., 10.X.2022, No202200747. Paratype: 1♀, Anhui, Langya Mountain, Yuanyuan Han, 10.X.2022, No202200755; 1♀, Fujian, Sanming, Xianbing Bai, 4.IX.2021, No202206759.

Description: Female (Figure 5) holotype. Body length 3.4 mm, fore wing length 2.8 mm.

Head: Antenna with 26 flagellomeres; first flagellomere ca. 3.4× longer than its apical width; face (Figure 6E) granulose, somewhat rugulose on the lower part; clypeus very weakly separate from face, granulose-punctate and becoming punctate apically; malar space mat, 0.67× as long as basal width of mandible; upper tooth of mandible equal to the length of lower tooth, with a weak lamella; frons granulose to granulose-punctate; vertex (Figure 6F) granulose; ocular-ocellar distance 1.1× longer than diameter of lateral ocellus, distance between lateral ocelli 1.4× longer than diameter of lateral ocellus; occipital carina complete, reaching hypostomal carina before base of mandible.

Mesosoma: Pronotum granulose-punctate dorsally and transversely striate ventrally. Mesoscutum (Figure 6G) granulose-punctate. Scutellum convex in profile, with lateral carinae basally, shallowly punctate. Mesopleuron (Figure 6B) punctate, trans-striate below tegula, speculum smooth and shiny, sternaulus present at basal 0.15; metapleuron punctate, slightly rugulose on juxtacoxal area, short rugose posteriorly; propodeum (Figure 6C and Figure 7C) with area basalis trapezoid; area superomedia hexagonal, short rugulose, 1.5× as long as wide; area petiolaris irregularly rugose, not concave medially, confluent with area superomedia; area externa subpolished and shallowly punctate with irregular rugulosity; area dentipara rugose to rugose-punctate; area lateralis subpolished, with shallow punctures; costulae missing; lateral longitudinal carina developed; propodeal spiracle oval.

Wing: Fore wing (Figure 6A) areolet present emitting vein 2m-cu from its apical half, with a moderate stalk. Marginal cell short, distal part of surrounding vein 2.2× longer than proximal one. Vein 1cu-a opposite M&RS. External angles of second discal cell acute (70°). Hind wing with CU&cu-a vertical, not intercepted.

**Figure 5 insects-15-00932-f005:**
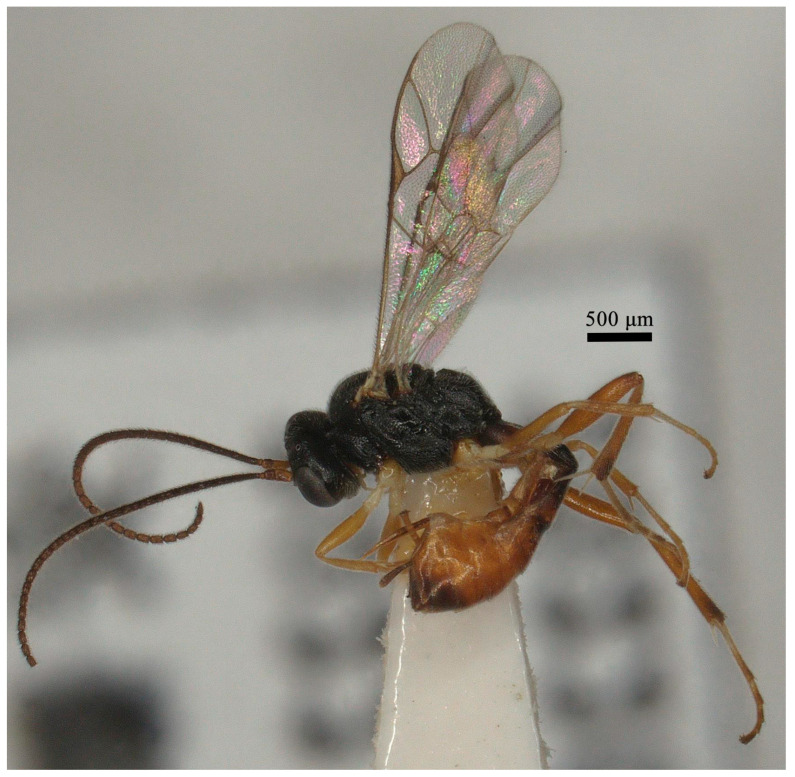
*Lemophagus nanus* sp. nov., holotype, female, habitus.

Legs: Hind femur ca. 5.0× longer than wide. First tarsal segment without ventral row of setae medially. Tarsal claws curved apically, weakly pectinate basally.

Metasoma: First metasomal segment (Figure 6H) without glymma, dorsolateral carinae of first tergite moderately present. First tergite 3.0× longer than width of postpetiole. Second tergite 0.7× as long as first tergite, 1.3× longer than its apical width; thyridium almost circular and separated from base of second tergite by length of its diameter. Third tergite 0.9× as long as its apical width. Ovipositor (Figure 6D) nearly as long as apical depth of metasoma, with a notch on upper valve.

Color: Black. Scape and pedicel yellowish-brown; mandible yellowish-brown except teeth reddish-brown apically; tegulae whitish-yellow; fore and mid coxae whitish-yellow, trochanter and trochantelli whitish-yellow, femur and tibia yellowish-brownish, first to fourth tarsi whitish except brownish apically and apical tarsi blackish-brown; hind coxae blackish-brown basally and gradually becoming yellowish apically, trochanter and trochantelli whitish-yellow, tibia apically and each tarsi apically blackish, remainder of hind leg yellowish-brown; first metasomal segment, basal 0.6 of second tergite, basal 0.2 of third tergite and metasoma dorso-apically blackish, remainder of metasoma yellowish-brown.

Distribution: China (Anhui, Fujian).

Comparative diagnosis: This species is similar to *L. japonicus* (Sonan, 1930), but differs from the latter by having a face that is granulose and somewhat rugulose in the lower part (rugulose-punctate in *L. japonicus*), a malar space 0.67× as long as the basal width of the mandible (as long as the basal width of the mandible in *L. japonicus*), a mesoscutum not rugose-punctate on the notaulic region (mesoscutum rugose-punctate on notaulic region) and an area superomedia with its lateral sides nearly parallel (area superomedia with its lateral sides convergent in *L. japonicus*).

The new species is also similar to *L. curtus* Townes, 1965, and *L. diversae*, Kusigemati, 1972, but differs from the latter two species by having a face that is granulose and somewhat rugulose in the lower part (rugose-punctate in *L. curtus* and *L. diversae*), an area superomedia with its lateral sides nearly parallel (convergent in *L. curtus* and *L. diversae*), missing costulae (costulae present in *L. curtus* and *L. diversae*) and fore and mid coxae that are whitish-yellow (blackish-brown in *L. curtus* and *L. diversae*).

Etymology. The specific epithet is the masculine form of the Latin adjective *nanus*, *-a, -um*, meaning small, referring to its small body.

**Figure 6 insects-15-00932-f006:**
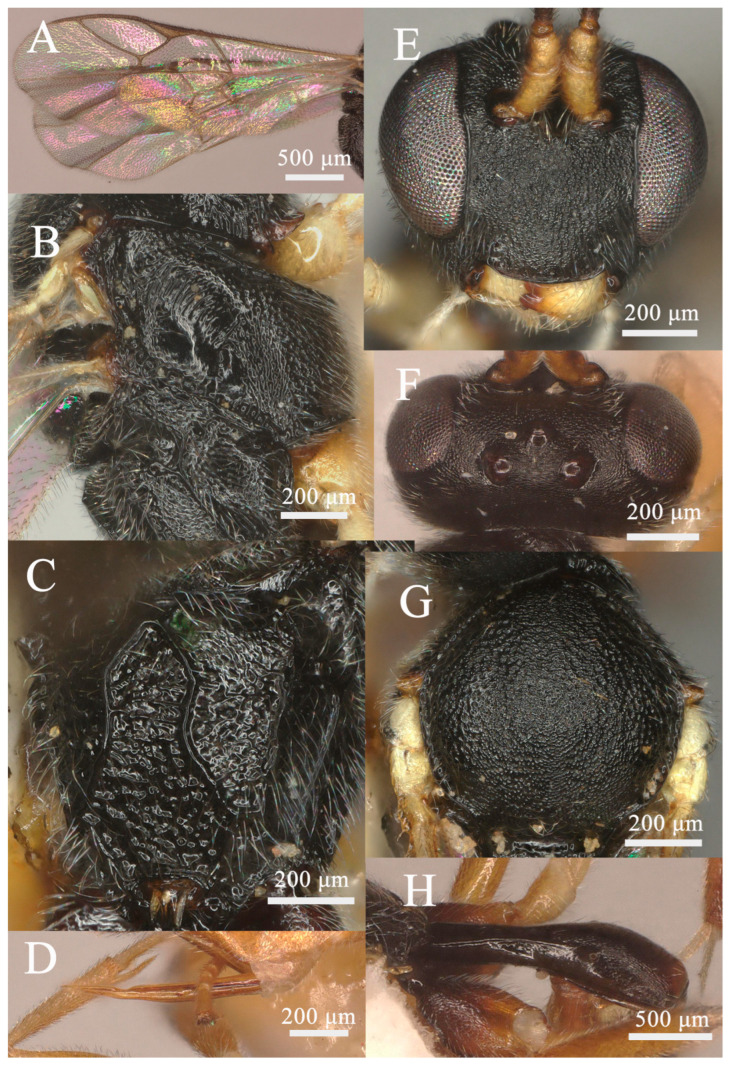
*Lemophagus nanus* sp. nov., holotype, female. (**A**) Fore wing; (**B**) Mesosoma, lateral view; (**C**) Propodeum, dorsal view; (**D**) Ovipositor, lateral view; (**E**) Head, anterior view; (**F**) Head, dorsal view; (**G**) Mesoscutum, dorsal view; (**H**) First metasomal segment, lateral view.

## 4. Discussion

Researchers have long been interested in the discovery of new distribution locations and previous unknown species as it reveals their diversity and evolution. Prior to this study, nine valid species were known all over the world. However, only one of them is from the Oriental region. In this study, one new species, *Lemophagus nanus* sp. nov., and one new record, for *L. curtus*, are reported for the first time from Oriental region, which expands the distribution range of the genus. Vas first reported *L. curtus* from North Korea, indicating that this species is widely distributed in the Palaearctic region [18]. Thus, our finding of *L. pulcher* collected from Eastern Palaearctic is not surprising, even though this species is generally reported from Europe. Their hosts are usually regarded as important agricultural pests distributed worldwide, which may explain the trans-Palaearctic distribution of *L. curtus* and *L. pulcher*.

The following characteristics are mainly used to distinguish different species for various authors: the proportion between the ocello-ocular distance and the diameter of the laterocellus; number of antennal segments; hind tibia with a unicolor or not; area superomedia open or closed behind and dorsolateral carinae of tergite 1 distinct or not [24,37]. In this study, we found some other characteristics useful for the identification of females of the *Lemophagus* species: a propodeum with costulae present, weak or absent; tarsal claws simple or strongly pectinate; and the coloration pattern of the metasoma.

**Figure 7 insects-15-00932-f007:**
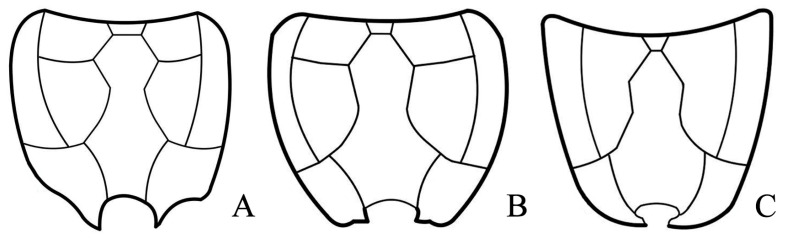
Propodeal carination (surface sculpture not indicated). (**A**) *L. curtus* Townes, 1965; (**B**) *L. pulcher* (Szépligeti, 1916); (**C**) *L. nanus* sp. nov.

## Data Availability

All data are available in the paper.

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
