# Peer review of "New Species and Records of *Lemophagus* Townes, 1965 (Hymenoptera, Ichneumonidae, Campopleginae), from China [Author-notes fn1-insects-15-00932]"

_insects, 2024, doi:10.3390/insects15120932_

Round 1
Reviewer 1 Report
Comments and Suggestions for Authors
Lemophagus has been rather neglected so it is nice to see some new species being described, from outside of the known range of the genus. Although the descriptions are mostly good I have some reservations about the validity and identification of some of the new species.
In the introduction it is stated that Lemophagus are ‘well characterized and can be easily differentiated from other genus…’ In fact, Lemophagus are very similar to some Olesicampe and identification can be problematic. There is no reference in this manuscript to the paper by Klopfstein et al. (2022), in which Lemophagus is diagnosed and an interactive key to European Campopleginae genera is provided. Klopfstein et al. (2022) also note that L. foersteri, supposedly reared from Spodoptera, is anomalous within the genus and that the best character for distinguishing Lemophagus from similar genera is the sculpture of the face, which is not mentioned here.
The new species are not differentiated from European species. Lemophagus ‘levipectus’ looks very similar to L. pulcher. Why is it not L. pulcher? And I am not convinced that L. longiantennus is actually a Lemophagus species. The clypeus looks too narrow so this really needs to be convincingly separated from Olesicampe of the ‘Holocremnus’ group.
The cited literature is not really up to date here. As well as missing Klopfstein et al. (2022), there is a recent body of work on Lemophagus species other than L. curtus being assessed for their biocontrol potential, particularly against the lily beetle, Liliocris lilii (e.g., Haye & Kenis, 2004).
The species descriptions include characters common to all Lemophagus (e.g., nervellus not intercepted, hypostomal and occipital carinae meeting distant from the mandible base, etc.) and can be trimmed down.
There is no indication of how Lemophagus curtus were identified and no illustrations, unlike the newly described species, so it is impossible to assess the accuracy of the identification.
I have made a few corrections and comments on the attached PDF. I didn't thoroughly check all the descriptions so it would be worth checking whether the corrections I made need to be implemented in all the descriptions.
Haye T, Kenis M (2004) Biology of Lilioceris spp. (Coleoptera: Chrysomelidae) and their parasitoids in Europe. Biological Control 29 (3):399-408
Klopfstein S, Broad GR, Urfer K, Vårdal H, Haraldseide H (2022) An interactive key to the European genera of Campopleginae (Hymenoptera, Ichneumonidae) and 20 new species for Sweden. Entomologisk Tidskrift 143 (3):121-156

Only minor corrections needed, as indicated on the review copy of the manuscript.
Reviewer 2 Report
Comments and Suggestions for Authors
Dear Authors and Editors,
I provide a brief summary here, and a more detailed evaluation below, with helpful suggestions.
Summary:
Sadly, this manuscript contains serious taxonomical and biogeographical errors, and must not be published in this present form. Two of the species proposed as new (L. areolatus and L. longiantennus) actually clearly does not belong to Lemophagus, and one of them (L. levipectus) would be a junior synonym of L. amurensis Kasparyan and Dbar, 1985 if described – just to mention the most fatal problems. However, there are valuable parts of the manuscript which, with considerable corrections and improvements, will be suitable for publication, after a quite major revision. Therefore, I chose to recommend major revision instead of rejection.
Taxonomical issues:
Since the authors apparently failed to understand the generic delimitations of Lemophagus, two of the species proposed as new (L. areolatus and L. longiantennus) actually clearly does not belong to Lemophagus, therefore must be removed from the revised version.
The source of the problems lies here (lines 32–35) „This genus is well characterized and can be easily differentiated from other genus by hind wing with nervellus not intercepted; glymma on petiole faintly indicated or entirely absent and ovipositor about as long as apical depth of metasoma.”
Lemophagus is not at all easily, but quite hardly distinguishable from the related genera such as Hyposoter and the formerly recognised “Holocremnus” group of Olesicampe. The reason is, that these morphologically defined genera are intermingled, as already pointed out by Horstmann (2004), noting that Lemophagus is probably a lineage of Hyposoter (al well as some groups of Olesicampe, see Galsworthy et al. 2023). The correct identification of the presently accepted Lemophagus can only be learned by examining several comparative materials, especially types (which is clearly missing in this study), not only those of Lemophagus species but also those of Hyposoter and Olesicampe, too. The characteristics the authors list in lines 33–35 are word by word true for several other genera, including Hyposoter and Olesicampe. The authors failed to notice or understand the most important characteristic combination of the genus: the clypeus is conspicuously wide, apical margin slightly out-turned and proposed AND the ovipositor with dorsal notch unusually removed from the apex (situated at or proximad to apical third). Both traits can be clearly examined in the uploaded photos of the holotype of L. amurensis. Horstmann’s revision (2004) of the genus seems to be neglected by the authors, although he emphasised the shape of the clypeus, which is one of the two key features to avoid the wrong generic identification. The shape of the clypeus is not even mentioned in the more detailed taxonomical diagnosis (lines 98–106), and its margin is wrongly interpreted by the authors as “blunt”. It is not blunt, see for yourself in your specimens of L. levipectus = L. amurensis, and in the uploaded photos of L. amurensis holotype.
With the actually accepted delimitation of the genus detailed above, the authors also must now see that neither L. areolatus and nor L. longiantennus belong to Lemophagus. They are definitely not. Both are either Hyposoter or the formerly recognised “Holocremnus” group of Olesicampe. I cannot tell for sure which one based on the photos for L. areolatus, because Fig. 2E show the unimportant upper face and frons instead of the important clypeus, but I may think it is probably Olesicampe. L. longiantennus is a Hyposoter, without doubt.
Please, mark my advice, and examine more materials, especially many types of the genera complex, to avoid another case similar to Picacharops: the same first author with different co-authors described a new Picacharops species in 2022, which I had to made nomina dubia in 2023 as it is clearly not a Picacharops. The authors would know if would relied on examination of real Picacharops specimens instead of relying only on texts of original descriptions. At least now in this Lemophagus case I could help to eradicate the embarrassing errors of genus-level misidentification in a species description before it is published. It is better for everyone interested.
L. levipectus is indeed a Lemophagus, but I think it is conspecific with L. amurensis Kasparyan and Dbar, 1985, a species known from eastern Russia. So it is a new record for China, it is expected to occur there. Worth to publish, but as a new record, not as a new species. As a courtesy, I have uploaded the photos of the holotype of L. amurensis for the authors. They clearly missed to examine it before, and nearly described a junior synonym of this species as L. levipectus. I cannot emphasize enough for the authors that reading the original descriptions is not enough in most cases, and the type materials must be examined to ensure good taxonomic choices, especially in the unusually difficult genera of Campopleginae. The photos are also helpful for the authors to understand the most important distinguishing characters of the genus Lemophagus: the shape of clypeus and the ovipositor with dorsal notch unusually removed from the apex. By checking the holotype of L. amurensis, their conspecificity is quite convincing. As far as I can see, there are only minor colour differences, which are not bases for a reliable species description. Note that Fig. 3. and Fig 4. is said to depict the same “holotype” specimen, however Fig. 4D is clearly not the ovipositor of the specimen of Fig. 3, the exposure and angle is different; something is mixed up here, must be corrected.
L. nanus is also a Lemophagus, maybe a new species, but I think it is conspicuously similar to Lemophagus diversae Kusigemati, 1972, which species is not even mentioned in the manuscript, although known from east Russia and Japan, and expected to occur in China! May be easily conspecific! The author must compare their material to the type material of L. diversae to be able to decide whether they are conspecific or not. I only have bad quality photo of the holotype of L. diversae, but I uploaded it for the authors. As you will see, very-very similar, maybe the same. Also, the distinction from L. japonicus (Sonan, 1930) in the comparative diagnosis is poor and not convincing: here (lines 357–361) the authors repeat only the characteristics of L. nanus, but not those of L. japonicus. I wonder why? Have they seen any actual specimen, not to mention type specimen of L. japonicus?
L. nanus must be compared with and clearly distinguished from L. crioceritor Aubert, 1986, too, because they also seem similar; L. nanus keys out in Horstmann”s key (2024) as L. crioceritor. If L. curtus is present from Western Europe to Korea, L. crioceritor can also be a transpalaearctic species, so it cannot be neglected like in the present form in the manuscript. I myself reported several European species for the first time from Mongolia and North Korea, it is common that many species are present in the whole Palaearctic region! So more work is needed here to reliably decide whether this represent a new record (of most probably L. diversae I think) from China or really a new species.
Biogeographical issues:
The biogeographical treatment is also erroneous. Townes in his Eastern Palaearctic catalogue (1965) clearly delimited the border between the Eastern Palaearctic and Oriental regions. Since that, all major catalogues of Ichneumonidae (such as Yu and Horstmann’ world checklist in 1999 and Taxapad Ichneumonoidea database catalogues) follow this delimitation. The authors should be aware of these basic works of ichneumonology and follow that too, otherwise they ruin the catalogue data. In this interpretation, area above 30°N is Eastern Palaearctic, below is Oriental. I have checked the localities in the manuscript, and found that L. levipectus and L. longiantennus represent Eastern Palaearctic species, not Oriental species! L. nanus and L. curtus represent both regions. So, in the revised version, they (L. levipectus = L amurensis, and L. nanus if not conspecific with L. diversae/crioceritor/japonicus) must be correctly treated as entirely or partly Eastern Palaearctic species. Naturally, this delimitation is arbitrary, but since all of our biogeographical knowledge of Ichneumonidae rely on this delimitation of regions, it must be followed.
Illustrations:
The manuscript is well illustrated, but sometimes only apparently so. In some cases they are entirely useless. As I mentioned, Fig. 2E show the unimportant upper face and frons instead of the important clypeus. In photos of frontal view of head, the clypeus must be clear. Figs 4C and 6C shows the very important propodeum in useless angles. I understand that in some cases it is impossible to make good photos of the propodeum due to the mounting of the specimen. At least in this cases, schematic drawing are to be made, these are often more straightforward than photos. As I mentioned, Fig. 3. and Fig 4. is said to depict the same “holotype” specimen, however Fig. 4D is clearly not the ovipositor of the specimen of Fig. 3, the exposure and angle is different; something is mixed up here, must be corrected.
Some other observations:
The description of area superomedia is inadequate in the descriptions. It is important, its shape, length/width etc. must be described with more details (see either Horstmann’s or mine descriptions). In the cases of Figs 4C and 6C, which are useless in this regard, the reader has no idea about one of the most important characters! It is not coincidence that most taxonomist of Campopleginae give schematic drawings of the propodeal carination, it is needed. More useful than photos which do not show the important features because of useless angles.
There are some other smaller errors in the text. For the correct use of terms of surface sculpture, the authors must study Harris (1979), because sometimes I disagree with their terminology when I have compared the text to the figures. I have uploaded the publication for the authors. In addition, some wrong morphological terms are present such as “area external” instead of are externa. Should be critically revised and improved. Also errors in the key, but I will not correct it, as it will be considerably changed due to the correction of the wrong taxonomical decision, or even may be unnecessary.
The “Discussion” is unnecessary in this paper. It is repetitive, not more than a longer abstract, there is nothing new in it. I recommend deleting it, not needed in my opinion.
Title: I recommend never call a paper as “review” in the title if the type material of all mentioned species were not examined. If they were not, it is rather “overview” than taxonomical “review”. No one can really review anything without checking all the relevant types. If this “review” term is used without complete type investigations, that gives a bad impression to taxonomists.
Etymologies: Always give grammatic description in the etimology, whether you consider the specific epithet as adjective or noun etc. In modern taxonomical works it is expected (see examples of phrasing in my papers).
Lines 22–23: There is no use and it is misleading to quote exact but outdated number of taxa, as many species and some genera have been described since Taxapad 2016. You can say that about 2200 species in more than 60 genera, no problem with that.
Final words:
I am very sorry for the opinion I had to give about this manuscript. Trust me I have worked a lot to be sure in my evaluation, and offered as much help as I can with type photos, etc. to protect you from the embarrassing genus-level misidentifications and descriptions of junior synonyms. I helped you even before the submission when you asked me to identify L. curtus for you. (Too bad you have not asked about the other, misidentified ones before submission…) I think you are a good student in ichneumonology, only your methods should be more improved to focus more on the direct examination of type and other comparative materials instead of rely mostly only on written texts. It is slower, harder sometimes, but helps a lot to avoid mistakes. I hope my comments can be used to improve your methods, both in the current major revision and future works. You are free to contact and ask me. I am willing to check the revised version, too.
With my regards,
Dr. Zoltan Vas
Hungarian Natural History Museum

Comments on the Quality of English LanguageOnly some minor corrections needed, no great problem here. E.g., formulae "something 0.84x longer than something" is incorrect, the correct form is "something 0.84x as long as something".
Round 2
Reviewer 2 Report
Comments and Suggestions for Authors
Dear Editors ans Authors,
I am generally satisfied with the revised version, much better than the previous one, and have some further, smaller concerns and recommendations; therefore, I decided that the manuscript needs some minor revisions.
The authors accepted, incorporated and answered the major problems pointed out in the previous review round. The wrongly identifed "Lemophagus" (actually Hyposoter and Olesicampe) specimens are correctly removed from the manuscript. As for the case of "L. levipectus"/L. amurensis/L. pulcher, I accept the authors choose the identification of the formerly proposed "L. levipectus" as L. pulcher. I have had personal communications with the first author in this regard after my review, and I pointed out to him that I think L. amurensis is a junior synonym of L. pulcher (they only show minor, non-diagnostic differences), so until I will have the possibility to personally examine the types of L. amurensis in St. Petersburg to synonymise this species with L pulcher, it is also acceptable for me to consider the Chinese specimens as L. pulcher, which is the senior synoym of L. amurensis in my evaluation, so eventually L. amurensis will also be validly called L. pucher.
Here I take to opportunity to recommend the authors to be aware that body size, minor surface sculpture differences and colouration differences in fore and middle coxae are at most weak indicators, but no diagnostic differences in Campopleginae. Body size is strongly determied by host quality (larval food) in ichneumonids, therefore shows high intraspecific variation. The colouration of fore and middle coxae are not nearly as stable as that of hind coxae. If more specimens are known from a species, these tend to show considerable variations, especially if these are not entirely light or entirely black but showing various intermedate forms. If only a few specimens are known of a species, just like in their present case, the authors should assume the same level of intraspecific variations which are known in better known relatives. In your current case, the L. curtus as a good example. The number of flagellomeres very rarely has any taxonomic, distinguishing value in Campopleginae as well. The development of propodeal carinae are good, stable, distinguishing characters in most campoplegines, but not always! In some genera these carinae are less stable and show unusual level of intraspecific variations - such as in Leptoperilissus, and yes, Lemophagus and Olesicampe, and, in lesser extent in Hyposoter. For example, the area superomedia of L. pulcher varies within the species (within the type series!) from posteriorly closed to opened. This alone should give a hint that propdeal carination are not as stable in this genus as in most other genera. These general, wider knowledge and insight are what I miss a bit from the work of the authors; these would also be needed to avoid the serious genus-level misidentifications of the previous version of the manuscript. For all the above, I recommend the authors while revising again the manuscript, do not over-emphasise traits like size, weak presence or absence of certain propodeal carinae, colouration of fore and middle coxae - neither in the key as distinguishing features nor in the diagnoses. Never forget that in taxonomy we classify specimens to species, not the other way around!
As for L. nanus, I have my concerns, I am not fully conviced as the adequate type comaparisons with L. diversae, L. japonicus and L. crioceritor are still missing (only bad, barely useful photos for diversae and japonicus, and nothing for crioceritor). I understand the authors were facing a deadline to submit the revised version, and it was not enough time to wait for the requested scientific-quality type photos (I know it takes weeks!). But I do not think it is an approppriate excuse to explain why doing poorer taxonomy than it should be. If I would have been in the present situation of the authors/editors, I would wait for these type photos, holding back the manuscript for the time necessary to examine the types in good enough photos at least. This is what I would do. I can accept if they choose to risk a synonym status rather than the adequate and safe, but more time-consuming method, because it is their taxonomic credibility they are risking. I am not saying that their taxonomic decision of describing L. nanus is wrong, probably it is good, only I can say that I am not fully convinced without the adequate examination of types of the similar species. I would not describe a new species without it, but I let the authors and editors decide this case. But as a minimum, the diagnosis of L. nanus must be extended and improved with L. diversae and L. crioceritor, too, indicating that the types of these similar species have not been adequatly studied and the given diagnostic information is based on the texts of the original descriptions. This is needed to indicate for the future reseacher to be informed that some risk is involved here.
Do not indicate in the key that L. pulcher is 9.0-9.4 long as its type specimens are ca 7 mm! this difference is within intrapecific variation, no problem, but if you use size as characters, use adequately and incorporate it already known range. Also, as I mentied above, minimitze the use of body lenght as distinguishing character to the smallest necessary level during the revision of the manuscript.
End of genus diagnosis in Inroduction / paragraph 2: "unusally far from apex" instead of "usually far from apex".
When I will get back the revised manuscript again after the present round, please provide a clear version without track changes, too, because it was very hard for me to read the revised ms full with track changes corrections.
I hope I have been helpful,
best wishes,
Dr. Zoltan Vas biologist, PhD, senior curator Head of Hymenoptera Collection Hungarian National Museum Public Collection Centre Hungarian Natural History Museum H-1088 Budapest, Baross u. 13. Hungary
Round 3
Reviewer 2 Report
Comments and Suggestions for Authors
Dear Authors and Editors,
the manuscript has improved a lot, thank you for your work. I have pointed out only a few remaining issues (some of them were previously pointed out by me, such as indication of body size ranges or more detailed diagnosis of the new species) which should be still improved, indicated by highlighted comments in the attached manuscript. Despite several attempt I was unable to upload the annotated pdf here, so I send it in e-mail attachment to the editor.
Best regards,
Zoltan
Dr. Zoltan Vas
Hungarian Natural History Museum
